# Mechanical and Structural Characterization of Laser-Cladded Medium-Entropy FeNiCr-B_4_C Coatings

**DOI:** 10.3390/ma16155479

**Published:** 2023-08-04

**Authors:** Artem Okulov, Yury Korobov, Alexander Stepchenkov, Aleksey Makarov, Olga Iusupova, Yulia Korkh, Tatyana Kuznetsova, Evgeny Kharanzhevskiy, Kun Liu

**Affiliations:** 1M.N. Mikheev Institute of Metal Physics, Ural Branch of the Russian Academy of Sciences, 620108 Ekaterinburg, Russia; 2Department of Physics, Ural Federal University, 620002 Ekaterinburg, Russia; 3Department of General Physics, Udmurt State University, 426034 Izhevsk, Russia; 4School of Materials Science and Engineering, Jiangsu University of Science and Technology, Zhenjiang 212100, China

**Keywords:** laser cladding, medium-entropy alloy, boron carbide, B_4_C, Raman spectroscopy, microstructure characterization, microhardness

## Abstract

Equiatomic medium-entropy alloy (MEA) FeNiCr-B_4_C (0, 1, and 3 wt.% B_4_C) coatings were deposited onto an AISI 1040 steel substrate using pulsed laser cladding. Based on an SEM microstructural analysis, it was found that the cross-sections of all the obtained specimens were characterized by an average coating thickness of 400 ± 20 μm, a sufficiently narrow (100 ± 20 μm) “coating–substrate” transition zone, and the presence of a small number of defects, including cracks and pores. An XRD analysis showed that the formed coatings consisted of a single face-centered cubic (FCC) γ-phase and the space group *Fm-3m*, regardless of the B_4_C content. However, additional TEM analysis of the FeNiCr coating with 3 wt.% B_4_C revealed a two-phase FCC structure consisting of grains (FCC-1 phase, *Fm-3m*) up to 1 µm in size and banded interlayers (FCC-2 phase, *Fm-3m*) between the grains. The grains were clean with a low density of dislocations. Raman spectroscopy confirmed the presence of B_4_C carbides inside the FeNiCr (1 and 3 wt.% B_4_C) coatings, as evidenced by detected peaks corresponding to amorphous carbon and peaks indicating the stretching of C-B-C chains. The mechanical characterization of the FeNiCr-B_4_C coatings specified that additions of 1 and 3 wt.% B_4_C resulted in a notable increase in microhardness of 16% and 38%, respectively, with a slight decrease in ductility of 4% and 10%, respectively, compared to the B_4_C-free FeNiCr coating. Thus, the B_4_C addition can be considered a promising method for strengthening laser-cladded MEA FeNiCr-B_4_C coatings.

## 1. Introduction

Medium- and high-entropy alloys (M-HEAs) are characterized by a combination of high ductility and microhardness values, making them very promising for a number of functional applications [1,2]. However, achieving these properties often requires the use of expensive components such as Ti, Nb, Zr, Hf, etc. [2,3,4]. The typical method to increase the hardness of traditional alloys is the formation of carbide, boride, and intermetallic phases in the material microstructure [5,6,7,8,9,10]. Boron carbide has a higher hardness (47 GPa) than most carbides and borides [11]. Moreover, B_4_C is characterized by thermodynamic stability and high wear and corrosion resistance [11,12], and its cost is 10–15 times lower than that of Nb, W, and Ti carbides, which are also used as hardening elements in HEAs. In addition, the mechanical properties of the resulting coatings can be modified by controlling the melt crystallization rate depending on the method of coating formation, including, for example, laser cladding [8] and arc spraying [13]. Alloying with carbides also makes it possible to control the ratio of primary and secondary carbides, which impacts both the hardness and ductility of the formed coatings [6].

The FeNiCr composition is the foundation for a significant number of M-HEAs [1,3,14,15,16]. In addition, it can also be considered an independent structural material with high corrosion resistance and promising mechanical characteristics [17,18,19,20,21,22]. It exhibits significantly enhanced wear resistance at elevated temperatures [23] and is low cost compared to other M-HEAs. According to [20], FeNiCr-based materials are commonly used as in-core components for nuclear light water reactors, as well as turbine disks and gas compressors. The mechanical and structural properties of FeNiCr materials under various conditions; for example, deformation temperatures, have been studied in [22,23,24,25,26]. In particular, as reported in [23], the MEA FeNiCr microhardness values exhibit an almost linear reduction with increasing temperature, decreasing from 267 HV at 25 °C to 204 HV at 600 °C. These characteristics make FeNiCr a promising candidate as a basic structural material for various applications, including the development of strong and wear-resistant composite coatings.

The addition of various reinforcing elements [27] in the manufacture of composite coatings via laser cladding allows for the activation and control of complex reactions during the formation of the cladding layer [15,21,26]. Laser cladding is an additive manufacturing process (similar to selective laser melting) used to deposit coatings on various types of metal surfaces. The structural evolution of the coatings during laser cladding was studied in [28], revealing significant impacts on the quality of the deposited coatings based on the size, shape, and arrangement of particles, as well as the laser beam intensity. Moreover, laser cladding is characterized by high heating and cooling rates and minimal mixing of the base and deposited materials. This makes it possible to form equiaxed fine-grained structures in the deposited metal coating with a minimum heat-affected zone and high adhesion between the deposited layer and the base material [28]. The laser cladding method allows for the use of inexpensive and abundant grades of steels and alloys in the manufacturing of wear-resistant coatings on the surface of various industrial products. However, it should be taken into account that the properties of the coatings formed on the substrate are influenced by the substrate material itself, and preference should be given to a material with a chemical composition to that of the formed coating. The use of more dispersed spherical powder also makes it possible to achieve more precise control of the laser cladding process, as shown in [27].

Thus, the research goal is to evaluate the impact of B_4_C addition on the mechanical and structural characteristics of laser-cladded MEA FeNiCr-B_4_C coatings.

## 2. Materials and Methods

### 2.1. Material Design

Mechanical FeNiCr-B_4_C compounds were prepared by mixing a custom spherical equiatomic FeNiCr powder (Table 1, PJSC “Ashinsky Metallurgical Plant”, Asha, Russia) with various amounts (1 and 3 wt.%) of commercial irregular-shaped B_4_C powder (LLC IPK “Umex”, Ufa, Russia). The fractions of the FeNiCr and B_4_C powders were 50–150 µm and 3–5 µm, respectively. The FeNiCr powder contained a significant amount of carbon (0.37 wt.%), which resulted from the application of ferrochromium during powder synthesis to reduce manufacturing costs.

Double-pass laser cladding of the FeNiCr-B_4_C (B_4_C-free FeNiCr, FeNiCr + 1 wt.% B_4_C, and FeNiCr + 3 wt.% B_4_C) coatings was carried out onto an AISI 1040 steel substrate (Table 2). The double-pass mode was chosen to reduce the mixing of the coating layer with the substrate materials. The general dimensions of the specimens were 10 × 5 × 5 mm, with average coating thickness of 400 ± 20 μm.

The laser machine was equipped with a ytterbium fiber laser with a maximum average power of 50 W and a wavelength of 1.065 µm. Laser cladding of all the FeNiCr-B_4_C (0, 1, and 3 wt.% B_4_C) coatings was performed by pulses of 40 ns duration in a chamber with a controlled Ar atmosphere. It is known that a short-pulse processing reduces a heat impact on a substrate structure [29].

### 2.2. Characterization Methods

The microstructure and chemical composition of the FeNiCr-B_4_C samples were investigated using scanning electron microscope (SEM) Quanta 200 Pegasus (FEI Company, Eindhoven, Netherlands), coupled with energy-dispersive X-ray spectroscopy (EDS) and electron backscatter diffraction (EBSD) detectors, and transmission electron microscope (TEM) JEM-200CX (JEOL Ltd., Tokyo, Japan).

The XRD analysis of the B_4_C-free FeNiCr and FeNiCr + 3 wt.% B_4_C coatings was carried out by Shimadzu XRD-7000 diffractometer (Shimadzu Corporation, Tokyo, Japan) coupled with graphite monochromator using CuK_α_ radiation. The diffraction spectrum was recorded in the angular range 2Θ = 30–120° with the scanning step ∆Θ = 0.03° and the pulse accumulation for 2 s. The X-ray reflections were identified using the X’Pert HighScorePlus 3.0.5 (Malvern Panalytical, Malvern, UK).

Raman spectra were measured using Raman confocal microscope Confotec^®^ MR200 (SOL Instruments, Augsburg, Germany) with a laser excitation wavelength of 532 nm. The laser power directed toward the sample surfaces was 76 mW. The acquisition time was 20 s with 5–10 accumulations per spectral segment. Local heating of the surface under such conditions was negligible. The 40× objective (Olympus Corporation, Tokyo, Japan) with a numerical aperture of 0.75 with a confocal pinhole diameter of 100 μm was used for this study. The spectra were recorded using 1200 lines/mm diffraction grating and an electrically cooled charge-coupled detector.

The micromechanical characteristics of the formed FeNiCr-B_4_C coatings, such as indentation hardness ($H_{\mathrm{IT}}$) and contact elasticity modulus ($E^{*}$), were determined by the instrumental indentation method using NanoTest-600 mechanical testing system (Micro Materials Ltd., Wrexham, UK) with Berkovich indenter. The microindentation tests were carried out at loading/holding/unloading times of 20 s, 20 s, and 20 s, respectively, and the maximum load of 250 mN. The measurement error of microindentation characteristics from 10 measurements was determined by a standard deviation with the confidence probability of 0.95. The following parameters of material resistance against elastoplastic strain were calculated: the ratio of the indentation hardness to the contact elasticity modulus $H_{\mathrm{IT}}/E^{*}$ [30], the elastic recovery $R_{e}=\left(\left(h_{\max}-{\;h}_{p}\right)/h_{\max}\right)\times 100\%$ [31,32], the power ratio $H_{\mathrm{IT}}{}^{3}/E^{*}{}^{2}$ [33], and the plasticity index $\delta_{A}=1-\left(W_{e}/W_{t}\right)$ [34], where $h_{\max}$ is the maximum indentation depth, $h_{p}$ is the residual depth, $W_{e}$ is the elastic deformation energy, and $W_{t}$ is the total deformation energy.

## 3. Results and Discussion

### 3.1. Microstructure and Chemical Composition

Figure 1 shows the custom spherical FeNiCr powder used for laser cladding. The SEM analysis of the powder confirmed the fraction in the range of 50–150 μm and the chemical composition close to the equiatomic one, indicated in the manufacturer’s test certificates. The purity (99.6%) and the fraction (3–5 µm) of the commercial irregular-shaped B_4_C powder used in this study also met the manufacturer’s specifications.

After mechanical characterization of all the synthesized FeNiCr-B_4_C coatings, it was found that the B_4_C-free FeNiCr and FeNiCr + 1 wt.% B_4_C samples possessed insignificant differences in the microhardness values. In addition, the etched cross-sections of all the obtained samples contained a homogeneous surface texture without any distinctive features such as a number of defects, inclusions, phases, etc. Therefore, only the B_4_C-free FeNiCr and FeNiCr + 3 wt.% B_4_C samples were discussed below.

The general view of the B_4_C-free FeNiCr and FeNiCr + 3 wt.% B_4_C cross-sections etched with HNO_3_ is shown in Figure 2. According to Figure 2, the average thickness of both coatings was 400 ± 20 μm. The remelting zone after laser exposure is clearly observed in Figure 2a.

Elemental mapping of the cross-sectional “substrate-coating” zone for the B_4_C-free FeNiCr sample is shown in Figure 3. It clearly demonstrates the transition (remelting) zone between the substrate and the coating materials. In Figure 3a, there are great purple color intensities corresponding to the Fe mapping and the absence of green and yellow color intensities corresponding to the Ni and Cr mappings in the substrate zone. This confirms that the AISI 1040 steel substrate contains approximately 97–99% Fe. However, the decrease in purple color intensity and the homogeneous distribution of purple, green, and yellow color intensities are observed in the transition and coating zones. The transition zone is especially visible on the Fe mapping, as indicated by the arrows. The technological regime of laser cladding contributes to obtaining the uniform thickness of the coating with the homogeneous microstructure.

The cross-sections of all the obtained samples were characterized by sufficiently narrow (100 ± 20 μm) “coating-substrate” transition zones, as shown in Figure 3. This indicated that the FeNiCr-B_4_C coatings, after 120 μm height, consisted only of the FeNiCr and B_4_C materials without mixing with the substrate. All the formed coatings possessed the same and insignificant number of defects, including cracks and pores. This was probably due to the presence of 0.37 wt.% carbon and the coarse dispersion (50–150 μm) of the initial FeNiCr powder.

### 3.2. Mechanical Characterization

Mechanical characterization defined the notable increase in microhardness with the slight decrease in ductility of the B_4_C-enriched FeNiCr coatings compared to the B_4_C-free FeNiCr one. The indentation microhardness of the FeNiCr coatings with 1 and 3 wt.% B_4_C rose by 16 and 38%, respectively, while the contact elasticity modulus decreased by 4 and 10%, respectively; see Figure 4 and Table 3. In addition, the mechanical characterization graphs confirmed that the dimensions of the transition zones (100 ± 20 μm) were consistent with the SEM results; see Figure 3.

According to Table 3, the evolution of microindentation parameters was observed due to the addition of 1 and 3 wt.% B_4_C. In particular, the parameters $H_{\mathrm{IT}}/E^{*}$, $H_{\mathrm{IT}}{}^{3}/E^{*}{}^{2}$, and $R_{e}$ of the FeNiCr + 3 wt.% B_4_C coating increased by 44%, 245%, and 41%, respectively, compared to the B_4_C-free FeNiCr one. It is known from the literature [30,32] that the specific contact hardness $\;H_{\mathrm{IT}}/E^{*}$ and the elastic recovery $R_{e}$ parameters characterize the share of the elastic deformation in the total deformation during the indentation process, whereas the $H_{\mathrm{IT}}{}^{3}/E^{*}{}^{2}$ parameter is considered to be a characteristic of the material resistance to plastic deformation [33]. The increase in the $H_{\mathrm{IT}}/E^{*}$ and $H_{\mathrm{IT}}{}^{3}/E^{*}{}^{2}$ values indicated the FeNiCr + 3 wt.% B_4_C coating became stiffer and more wear-resistant. The observed increase in the $R_{e}$ parameter specified the improved ability of the FeNiCr + 3 wt.% B_4_C coating to elastically resist mechanical stress up to plastic deformation. The plasticity $\delta_{A}$ index, on the contrary, decreased by 9%, which was the result of the FeNiCr + 3 wt.% B_4_C coating’s hardening. This suggested that the FeNiCr-B_4_C coatings became less able to deform without fracturing under stress, which could also be due to a change in the chemical composition of the latter, namely, an increase in the carbon content.

According to [23], the microhardness value of the equiatomic MEA FeNiCr measured at room temperature was approximately 2.618 GPa, while the microhardness values of the FeNiCr-B_4_C (1 and 3 wt.% B_4_C) coatings, obtained in this study, were about 32 and 49% higher, respectively, compared to the reported alloy. Traditional research methods such as XRD and TEM analyses were used to interpret the improved mechanical behavior, in particular, to identify the distribution of B_4_C phases inside the synthesized FeNiCr-B_4_C coatings.

### 3.3. X-ray Diffraction Analysis

As mentioned earlier, due to the fact that the difference in mechanical properties between coatings without and with 1 wt.% B_4_C was insignificant, the XRD analysis was performed for the B_4_C-free FeNiCr and FeNiCr + 3 wt.% B_4_C coatings; see Figure 5. It was found that both coatings consisted of a single FCC γ-phase, space group *Fm-3m*. The intensity of the XRD spectra was distorted because the crystallites of both samples were textured. In particular, the integral intensity of the (111) Bragg peak, maximum in the untextured sample, was much less than the (200) peak.

The diffraction lines of both samples were significantly broadened, which indicated the defectiveness of the crystal structure (including the stressed state). The lines of the FeNiCr + 3 wt.% B_4_C coating were wider than the lines of the B_4_C-free FeNiCr sample (Figure 5). This indicated that the defectiveness (or stresses) was higher in the FeNiCr + 3 wt.% B_4_C coating.

Table 4 shows the phase composition with the calculated unit cell parameters of the formed B_4_C-free FeNiCr and FeNiCr + 3 wt.% B_4_C coatings.

According to the results, alloying with B_4_C did not impact the phase composition of the FeNiCr-B_4_C coatings. The addition of 3 wt.% B_4_C caused negligible lattice distortion: the increase in the interatomic distance and, accordingly, the lattice volume. The number of pores and cracks was not increased with the addition of 1 and 3 wt.% B_4_C.

### 3.4. TEM Analysis

TEM analysis was carried out for more detailed microstructural characterization of the present FeNiCr-B_4_C coatings, in particular, to identify B_4_C phases. Since the probability of detecting B_4_C phases increases with their higher concentration inside the coatings, the FeNiCr + 3 wt.% B_4_C sample was chosen for the TEM study. 

The microstructural characteristics of the FeNiCr + 3 wt.% B_4_C coating are shown in Figure 6. It was characterized by a two-phase FCC structure consisting of the grains (FCC-1 phase, *Fm-3m*) up to 1 µm in size and banded interlayers (FCC-2 phase, *Fm-3m*) between the grains (Figure 6a,b). The grains were clean with a low density of dislocations. All reflections in Figure 6b belonged to the FCC structure, which indicated the separation of two phases. In another section (Figure 6d), it can be seen that the FCC-2 phase was segregated in the form of separate rounded particles, generating a banded structure. The FCC-2 phase particles were formed predominantly along the grain boundaries of the FCC-1 phase and possessed a rounded shape at the beginning of nucleation, then elongated and grew to grains about 1 µm in size. The reflections of the FCC-2 phase sections had an elongated shape, which indicated that the observed bands inside the FCC-2 phase grains were planes located perpendicular to the foil plane.

The reflections that can be attributed to cementite were present in all the microdiffraction patterns. As shown in Figure 6a, the separate particle glows in the cementite reflection (002). The cementite particles were predominantly located inside the FCC-2 phase as separately highlighted with the dotted line in Figure 6a. The extended areas with only the FCC-1 phase are observed in Figure 6c. The following lattice parameters of the FCC-1 and FCC-2 phases were calculated based on the electron microscopy data: $a_{\mathrm{FCC}\text{-}1\;\mathrm{phase}}$ = 3.56 and $a_{\mathrm{FCC}\text{-}2\;\mathrm{phase}}$ = 3.50. The increase in the lattice parameter of the FCC-1 phase was presumably due to the enrichment in Cr. The observed two-phase FCC structure, formed upon the B_4_C addition, positively impacted mechanical characteristics, in particular, the hardening of the synthesized FeNiCr + 3 wt.% B_4_C coating. However, it was not possible to identify B_4_C phases, despite the use of sufficiently sensitive detection methods such as XRD and TEM.

### 3.5. Raman Spectroscopy Analysis

Raman spectroscopy was used as an additional highly-sensitive method to determine the molecular bonds of near-surface layers, in particular, the FeNiCr-B_4_C coatings. This is a qualitative method to identify light elements such as B, C, and, most importantly, B_4_C phases relevant to the present study.

The four characteristic surface areas named spots 1 (grey areas), 2 (white areas), 3 (dark areas), and microdrops were chosen on the surface of all the FeNiCr-B_4_C (0, 1, and 3 wt.% B_4_C) coatings for Raman spectra detection. Optical images of the FeNiCr-B_4_C (0, 1, and 3 wt.% B_4_C) coatings with the marked detection spots are shown in Figure 7. Examples of the Raman spectra measured on the AISI 1040 steel substrate and the described surface spots of the FeNiCr-B_4_C coatings are shown in Figure 8. Figure 8 shows the difference in intensity of the Raman bands between different detection spots. In order to accurately compare the positions of each Raman peak, averaged over 5–7 spectra, they were identified at each detection spot for all the samples. The interpretation of the Raman frequencies is presented in Table 5.

According to Figure 8, the Raman spectra detected in the B_4_C-free FeNiCr and FeNiCr-B_4_C coatings significantly differ from the spectra of the AISI 1040 steel substrate. The most notable Raman peaks of the AISI 1040 steel substrate corresponded to the α-Fe_2_O_3_ phase [35]. Metals such as Fe, Ni, and Cr, which were included in the composition of the coatings and not capable of producing Raman spectra, appeared in the form of metal oxides due to the interaction of the metal with oxygen. The formed FeNiCr-B_4_C (0, 1, and 3 wt.% B_4_C) coatings were characterized by low-intensity iron oxide (α-Fe_2_O_3_, γ-Fe_2_O_3_) peaks (Figure 8a,b). Among all the detection spots, it was found that spots 3 (Figure 8c) and the microdrops (Figure 8d) of the FeNiCr-B_4_C coatings were characterized by the absence of peaks (~206, ~270 cm^−1^), corresponding to the α-Fe_2_O_3_ phase in comparison with spots 1 and 2.

High-intensity peaks of 527–543 cm^−1^ and 657–672 cm^−1^ corresponding to the Cr_2_O_3_ and NiCr_2_O_4_ phases [36], respectively, were observed for all the FeNiCr-B_4_C coatings. In particular, spots 2 were characterized by increased NiCr_2_O_4_ phase content inside the FeNiCr-B_4_C (0 and 1 wt.% B_4_C) coatings (Figure 8b). The lower and broader peak of 657 cm^−1^, corresponding to the NiCr_2_O_4_ phase, indicated the ability of the FeNiCr + 3 wt.% B_4_C coating to exhibit greater in-plane tensile strain compared to the FeNiCr-B_4_C (0 and 1 wt.% B_4_C) coatings. In addition, this fact was confirmed by the shift of the peak maxima towards lower frequencies when the concentration of B_4_C increases.

The peaks of 469–478 cm^−1^ corresponding to the stretching mode of C-B-C chains in B_4_C carbides [37,38] and the high-intensity peaks of 1328–1330 cm^−1^, 1554–1560 cm^−1^ corresponding to amorphous carbon (D and G peaks), respectively, detected on the dark areas (spots 3) of the FeNiCr-B_4_C coatings (Figure 8c), are observed only for their Raman spectra. Moreover, the vibrational mode of B_11_C icosahedra [37] was also found in the Raman spectra of the FeNiCr-B_4_C (1 and 3 wt.% B_4_C) coatings. This confirmed the presence of B_4_C carbides in these samples.

Thus, Raman spectroscopy allowed us to identify B_4_C phases and to establish the reasons for the increased microhardness of the synthesized FeNiCr-B_4_C samples, especially for the FeNiCr + 3 wt.% B_4_C coating.

## 4. Conclusions

The study evaluated the impact of B_4_C addition on the mechanical and structural characteristics of laser-cladded MEA FeNiCr-B_4_C coatings. The following main results were obtained:The formed FeNiCr-B_4_C (0, 1, and 3 wt.% B_4_C) coatings were characterized by an equiaxed coarse-grained structure. According to XRD analysis, the coatings were homogeneous solid solutions and consisted of a single FCC γ-phase, space group *Fm-3m*, regardless of the B_4_C content. Additional TEM analysis of the FeNiCr coating with 3 wt.% B_4_C showed that it consisted of the grains (FCC-1 phase, *Fm-3m*) up to 1 µm in size and banded interlayers (FCC-2 phase, *Fm-3m*) between the grains. The grains were clean with a low density of dislocations.The cross-sections of all the obtained samples were characterized by an average coating thickness of 400 ± 20 μm and a sufficiently narrow (100 ± 20 μm) “coating-substrate” transition zone. This indicated that the coating, after 120 μm height, consisted only of the equiatomic FeNiCr powder material without mixing with the substrate due to the double-pass laser cladding.All the obtained coatings possessed the same and insignificant number of defects, including cracks and pores. This was probably due to the presence of 0.37 wt.% carbon and the coarse dispersion (50–150 μm) of the initial FeNiCr powder. Thus, it can be concluded that there was no impact of B_4_C addition on the defect formation.Raman spectroscopy confirmed the presence of B_4_C carbides in the FeNiCr + 1 wt.% B_4_C and FeNiCr + 3 wt.% B_4_C coatings by detected peaks corresponding to amorphous carbon and peaks indicating the stretching of C-B-C chains in B_4_C carbides.The mechanical characterization of the FeNiCr-B_4_C coatings specified that additions of 1 and 3 wt.% B_4_C resulted in a notable increase in microhardness of 16% and 38%, respectively, with a slight decrease in ductility of 4% and 10%, respectively, compared to the B_4_C-free FeNiCr coating. Thus, the B_4_C addition can be considered a promising method for strengthening laser-cladded MEA FeNiCr-B_4_C coatings.

## Figures and Tables

**Figure 1 materials-16-05479-f001:**
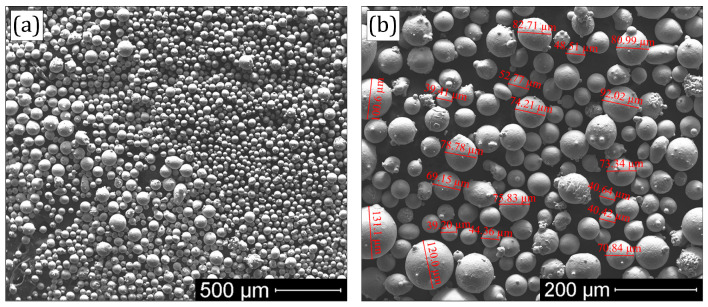
Morphology of the custom FeNiCr powder at (**a**) 100× and (**b**) 300× magnifications.

**Figure 2 materials-16-05479-f002:**
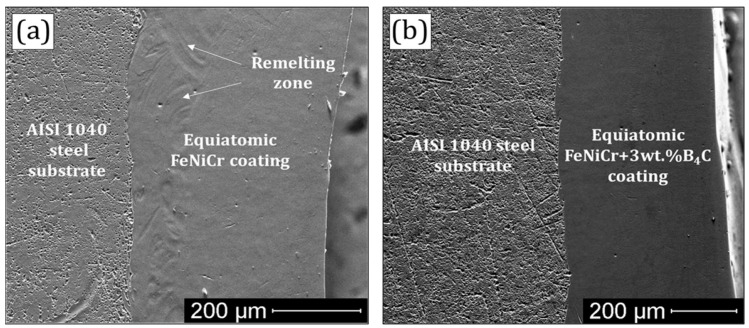
General view of the (**a**) B_4_C-free FeNiCr and (**b**) FeNiCr + 3 wt.% B_4_C cross-sections.

**Figure 3 materials-16-05479-f003:**
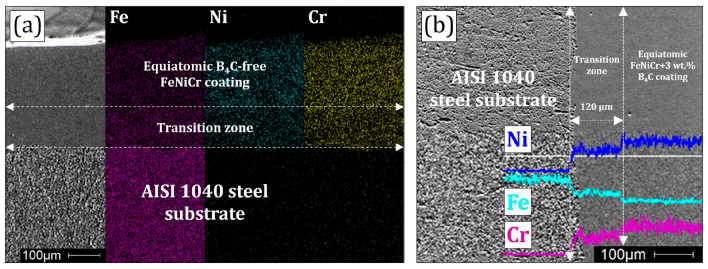
(**a**) Elemental mapping and (**b**) chemical analysis line of the cross-sectional “substrate-coating” zones of the B_4_C-free FeNiCr and FeNiCr + 3 wt.% B_4_C samples, respectively.

**Figure 4 materials-16-05479-f004:**
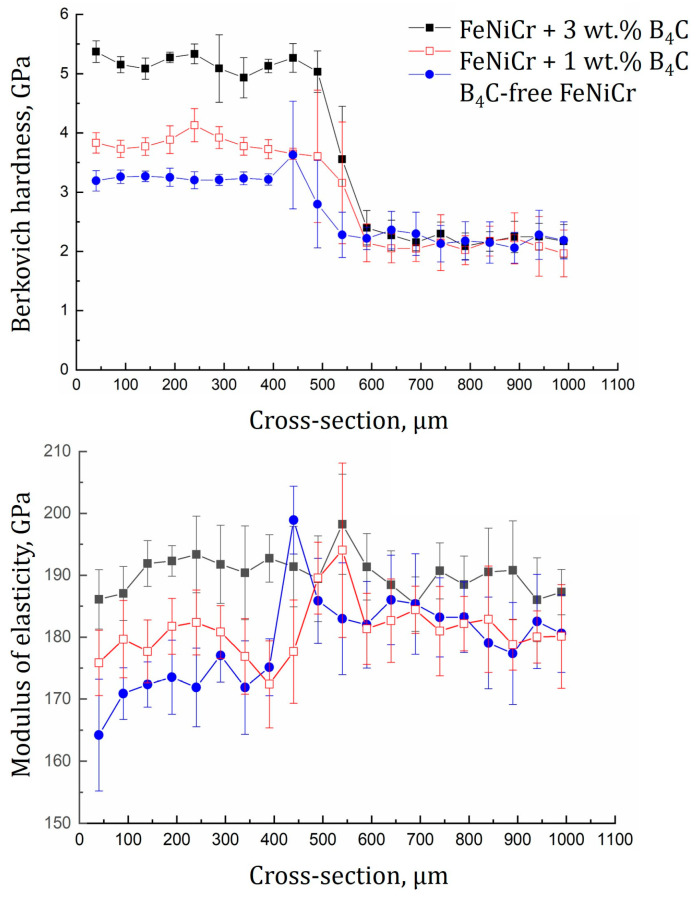
Mechanical characterization of the FeNiCr-B_4_C coatings.

**Figure 5 materials-16-05479-f005:**
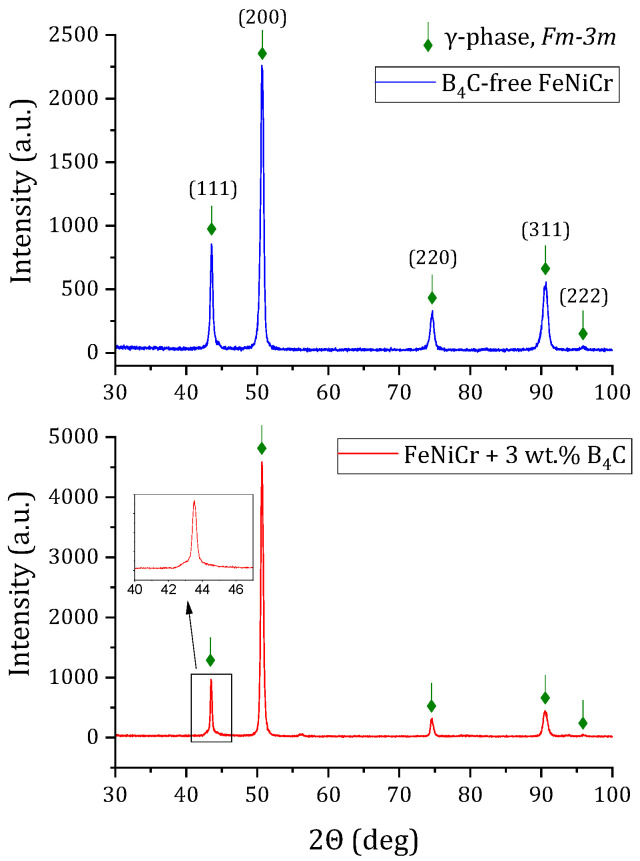
XRD patterns of the B_4_C-free FeNiCr and FeNiCr + 3 wt.% B_4_C coatings.

**Figure 6 materials-16-05479-f006:**
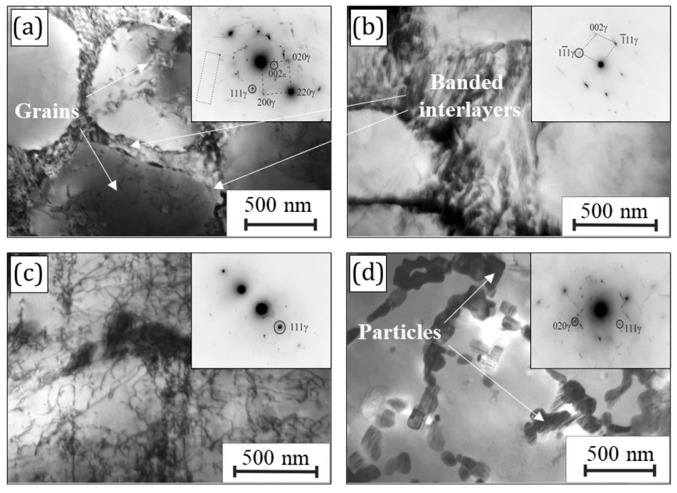
TEM images with diffraction patterns of the FeNiCr + 3 wt.% B_4_C coating: (**a**) grains, (**b**) banded interlayers, (**c**) FCC-1 phase structure, and (**d**) separate rounded particles.

**Figure 7 materials-16-05479-f007:**
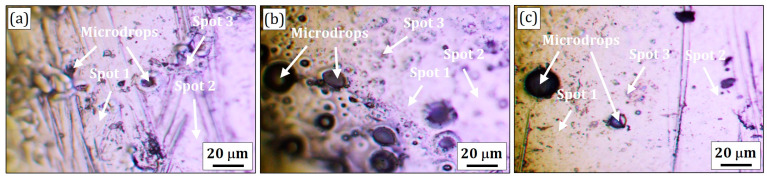
Optical images of the (**a**) B_4_C-free FeNiCr, (**b**) FeNiCr + 1 wt.% B_4_C, and (**c**) FeNiCr + 3 wt.% B_4_C coatings with marked detection spots of Raman spectra.

**Figure 8 materials-16-05479-f008:**
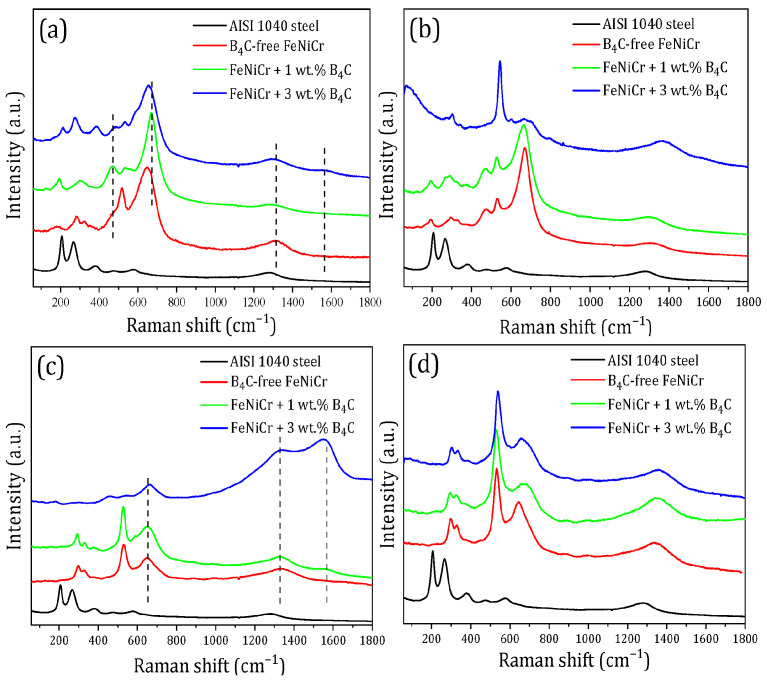
Raman spectra of the AISI 1040 steel substrate, B_4_C-free FeNiCr, FeNiCr + 1 wt.% B_4_C and FeNiCr + 3 wt.% B_4_C coatings: (**a**) spots 1, (**b**) spots 2, (**c**) spots 3, and (**d**) microdrops. The dashed lines visually identify the shift of Raman peaks for the FeNiCr-B_4_C coatings.

**Table 1 materials-16-05479-t001:** Chemical composition of equiatomic FeNiCr powder declared by the manufacturer (wt.%).

Fe	Ni	Cr	C	S	P	Si
Base	35.6	29.8	0.37	<0.001	0.008	1.62

**Table 2 materials-16-05479-t002:** Chemical composition of AISI 1040 steel (wt.%).

Fe	Mn	C	S	P	Si
98.6–99	0.6–0.9	0.37–0.44	≤0.05	≤0.04	0.15–0.35

**Table 3 materials-16-05479-t003:** Comparison of the FeNiCr-B_4_C coatings’ mechanical characteristics (variation (↓↑), %).

Sample	$H_{IT},\;\mathrm{GPa}$	$E^{*},\;\mathrm{Gpa}$	$H_{IT}/E^{*}$	$H_{IT}{}^{3}/E^{*}{}^{2},\;\mathrm{Gpa}$	$R_{e},\;\%$	$\delta_{A}$
B4C-free FeNiCr	3.228	172.11	0.0188	0.0011	5.9	0.86
FeNiCr + 1 wt.% B4C	3.825	178.34	0.0214	0.0018	6.6	0.83
FeNiCr + 3 wt.% B4C	5.167	190.64	0.0271	0.0038	8.3	0.78
The comparison of 0 and 3 wt.% B4C	60 ↑	11 ↑	44 ↑	245 ↑	41 ↑	9 ↓

**Table 4 materials-16-05479-t004:** Phase composition and unit cell parameters of the B_4_C-free FeNiCr and FeNiCr + 3 wt.% B_4_C coatings.

Sample	Phase	Unit Cell Parameters, Å	
		*a*	*V*
B_4_C-free FeNiCr	γ-phase, *Fm-3m*	3.598	46.58
FeNiCr + 3 wt.% B_4_C	γ-phase, *Fm-3m*	3.599	46.617

**Table 5 materials-16-05479-t005:** Average peak positions of Raman bands detected for different surface spots: AISI 1040 steel substrate, B_4_C-free FeNiCr, FeNiCr + 1 wt.% B_4_C, and FeNiCr + 3 wt.% B_4_C coatings.

Peak Position, cm^−1^				Interpretation
AISI 1040 Steel	B_4_C-Free FeNiCr	FeNiCr + 1 wt.% B_4_C	FeNiCr + 3 wt.% B_4_C	
207	196	194	209	α-Fe_2_O_3_
267	293	279	273	α-Fe_2_O_3_
380	-	377	383	γ-Fe_2_O_3_
-	-	469	478	Stretching of C-B-C chains in B_4_C
480	477	-	-	NiO
-	518	-	-	Fe_3_O_4_
-	-	533	532	Vibrational mode of B_11_C icosahedra
-	527	528	543	Cr_2_O_3_
578	-	-	-	γ-Fe_2_O_3_
645	648	-	-	γ-Fe_2_O_3_
-	672	665	657	NiCr_2_O_4_
1280	-	-	-	α-Fe_2_O_3_
-	1330	1328	1330	Amorphous carbon (D peak)
-	-	1560	1554	Amorphous carbon (G peak)

## Data Availability

Data will be made available on request.

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
