# Peer review of "Mechanical and Structural Characterization of Laser-Cladded Medium-Entropy FeNiCr-B_4_C Coatings"

_materials, 2023, doi:10.3390/ma16155479_

Round 1

Author Response

Following are my comments:

  1. This paper do not show any new knowledge.
  2. The research findings are not significant for publication.
  3. This papers lacks novelty in the area of laser cladding of composite coating on carbon steel.
  4. The conclusions are not having much impact in the subject area.
  5. The SEM figures are not showing any significant idea.
  6. This paper has been presented poorly and lots of grammatical mistakes are present.
  7. References are not cited properly
  8. What is the meaning of mechanical Fe-Ni-Cr-B4C compound?
  9. Overall, this paper is not suitable for publication in this journal.
  10. This paper should be rejected.

Answer: Thanks for your comments. They were taken into account during the revision of the manuscript.

Reviewer 2 Report

The paper deals with the characterization of FeNiCr – B4C coatings produced by laser cladding. The work is interesting, but some changes are required prior to publication.

1) Figure 1 is useless since it adds nothing to the aim of the paper. It only shows the dimension of few particles of a commercial powder.

2) The transition zone in Figure 3b (and in Figure 2b too!) is not evident from SEM image. In the same figure (3b) it is clear that from the substrate to the coating there is an increase of Ni and Cr and a decrease of Fe but what is the meaning of the white line in the middle of the image? Do the Ni, Cr and Fe graphs refer to any quantitative scale?

3) The authors state twice in the text that the differences in mechanical properties between coatings without and with 1 wt.% B4C are insignificant. So, why do they report the 1% data in the discussion of the mechanical properties and in Figure 4 (and in the discussion of Raman spectra)?

4) In Figure 5, the lines of the FeNiCr+3wt.%B4C coating do not seem to be noticeably wider than the lines of the sample.

5) In the Raman peaks of Figure 8a, attributed to NiCr2O4, the shift is not to lower frequencies with increasing B4C concentration. Indeed, the frequency of the B4C-free FeNiCr peak is lower than that of the FeNiCr+1wt.%B4C one, and those values clearly do not correspond to the values reported in Table 6 (it is not the only case). In general, the Raman section should be discussed in a clearer way since the text is quite confuse and the assignments should be reported on the spectra. In fact, the figures should be understandable also without the text.

Author Response

1) Figure 1 is useless since it adds nothing to the aim of the paper. It only shows the dimension of few particles of a commercial powder.

Answer: Thanks for your observation. However, this is a commercial powder manufactured specifically for our experiment. Moreover, one of the reviewers just asked us to add figures of the commercial powder in order to clearly see the powder morphology, in particular, sphericity and fraction.

2) The transition zone in Figure 3b (and in Figure 2b too!) is not evident from SEM image. In the same figure (3b) it is clear that from the substrate to the coating there is an increase of Ni and Cr and a decrease of Fe but what is the meaning of the white line in the middle of the image? Do the Ni, Cr and Fe graphs refer to any quantitative scale?

Answer: You are absolutely right that the transition zone in Figures 2b and 3b is not visible. For this goal, the chemical analysis line was built for the FeNiCr+3wt.%B4C sample. The white line in the middle of image 3b is the line that appears in the picture when the chemical analysis line is plotted using SEM (let's say this is a feature of the program and it cannot be removed). The Ni, Cr and Fe graphs reflect the quantitative ratio of these elements, thereby, visually showing the transition (remelting) zone of the «substrate-coating».

3) The authors state twice in the text that the differences in mechanical properties between coatings without and with 1 wt.% B4C are insignificant. So, why do they report the 1% data in the discussion of the mechanical properties and in Figure 4 (and in the discussion of Raman spectra)?

Answer: This is a fair comment. It should be noted that, first of all, it was carried out mechanical characterization of all three FeNiCr-B4C (0, 1 and 3 wt.% B4C) samples. Mechanical characterization showed that the change in mechanical characteristics between additions of 0 and 1% boron carbide is negligible. Further, after detailed characterization of the mechanical properties indicated in Table 4, our attention was focused on samples with 0 and 3% boron carbide. Thus, the study of the coatings’ structure by XRD, SEM and TEM analyzes continued on these samples.

Next, we are faced with the fact that such methods as XRD, SEM and TEM did not allow to establish the presence of boron carbide inside coatings. Therefore, in our work we use a highly sensitive method of Raman spectroscopy, which allows us to detect traces of oxides, carbides, etc. Thus, even a small addition of 1% boron carbide can be identified by the Raman spectra, which was shown in Figure 8. The main goal was to show the evolution of the intensity of the Raman spectra.

4) In Figure 5, the lines of the FeNiCr+3wt.%B4C coating do not seem to be noticeably wider than the lines of the sample.

Answer: We think you are right here. The word «noticeably» has been removed. Thank you for your comment.

5) In the Raman peaks of Figure 8a, attributed to NiCr2O4, the shift is not to lower frequencies with increasing B4C concentration. Indeed, the frequency of the B4C-free FeNiCr peak is lower than that of the FeNiCr+1wt.%B4C one, and those values clearly do not correspond to the values reported in Table 6 (it is not the only case). In general, the Raman section should be discussed in a clearer way since the text is quite confuse and the assignments should be reported on the spectra. In fact, the figures should be understandable also without the text.

Answer: Thank you for your observation. We agree with the poor illustration of the shift in the Raman peaks attributed to NiCr2O4 in Figure 8a. Fig. 8 was mainly intended to illustrate the difference in Raman bands intensity between surface spots (where is the spot with intense carbide bands, where is the spot with intense iron oxide peaks etc.) and only one spectrum from each sample has been shown as example for each spot without any peak positions assignments not to visually overload the figure. For proper specimen characterization we have measured about 5-7 Raman spectra at different points of each detection spots (white, violet, dark surface regions and microdrops) around the whole surface of each specimen. To thoroughly compare between the samples in Table 6 we have presented averaged values of each peak position among different detection spots for each specimen. We used only these averaged values to make a conclusion about tendency in shifts in the Raman peak positions due to boron carbides addition. That is why values in Figure 6 do not totally correspond with the Table 6 values.

For more clear visualization we have revised Figure 8a and replaced the spectrum corresponding to B4C-free FeNiCr with spectrum more correspondent to average values of NiCr2O4 peak position. Also, we have revised the description of Table 6.

Reviewer 3 Report

The manuscript is focused on the production of Equiatomic medium-entropy alloy FeNiCr- coatings by using laser cladding technique. There are several mistake throughout the text. There are several issues to clarify

INTRODUCTION

1-The introduction is not well written therefore it is hard to understand;

 for example:

row 40: ... to decrease in the plasticity of the latter...: what does it refer to?

row 64:... melt  pool is formed from the substrate and deposited (powder or wire) materials: what does it means?

2- real examples of application are not explicated

3- before use acronym it is necessary to explicate the meaning (es: SLM row 62)

4- the verb "to impact" should be used followed by "with"

RESULTS

5-TABLE 3: there is no carbon even if authors stated that an amount of carbon was presents in row 87

6-Figure 3. (a) Elemental mapping and (b) chemical analysis line of the cross-sectional “substrate- 168

coating” zones of the B4C-free FeNiCr and FeNiCr+3wt.%B4C samples, respectively.  From this caption I understand: Elemental mapping is on B4C-free FeNiCr sample. If it is correct why a similar image of FeNiCr+3wt.%B4C is not reported?

7-row 207: (0, 1 and 3 wt.% B4C), if I understand correctly, it should be  (1  and 3 wt.% B4C),

8- table 5: how authors calculated the reported parameters?

9- In my opinion ref 39 in  row 292 is not pertinent

10-as regard figure 7: it is not clear in which way authors analysed the same spot in all the sample

11- in Raman analysis they found oxides: it is necessary to discuss this issue

Extensive editing of English language are require, some suggestions are reported  in the revisions

Author Response

INTRODUCTION

1 - The introduction is not well written therefore it is hard to understand; for example:

row 40: ... to decrease in the plasticity of the latter...: what does it refer to?

Answer: Yes, you are completely right. We have removed the phrase «which in turn leads to decrease in the plasticity of the latter [7, since this phrase does not quite reflect the logic of the Introduction.

row 64:... melt  pool is formed from the substrate and deposited (powder or wire) materials: what does it means?

Answer: We agree with you, this text is really misleading. And since the above text, as in the first case, does not reflect the logic of the Introduction and, most importantly, it is not informative, we removed it. Thank you for your comment.

2 - real examples of application are not explicated

Answer: Thanks for your comment. We have added information about the possible use of our materials.

3 - before use acronym it is necessary to explicate the meaning (es: SLM row 62)

Answer: Yes, you are absolutely right. We took into account your comment and explicate this abbreviation in the text as «selective laser melting».

4 - the verb "to impact" should be used followed by "with"

Answer: Unfortunately, we did not find the verb "to impact" in the Introduction. If you meant this sentence «Thus, the research goal is to evaluate the impact of in-situ alloying with B4C on mechanostructural characteristics of laser deposited medium-entropy FeNiCr coatings.». The word «the impact» is used as a noun. But thanks a lot for your recommendation.

RESULTS

5 - TABLE 3: there is no carbon even if authors stated that an amount of carbon was presents in row 87

Answer: We agree, thanks for your observation. The main goal was to show the equiatomicity of the commercial powder. The EDS analysis also detected carbon in the amount declared by the manufacturer. We removed this table so as not to mislead the reader and left the information in the text.

6 - Figure 3. (a) Elemental mapping and (b) chemical analysis line of the cross-sectional “substrate-coating” zones of the B4C-free FeNiCr and FeNiCr+3wt.%B4C samples, respectively.  From this caption I understand: Elemental mapping is on B4C-free FeNiCr sample. If it is correct why a similar image of FeNiCr+3wt.%B4C is not reported?

Answer: Thank you for your comment. The main goal was to show the thickness of the transition (remelting) zone. This can be done both with elemental mapping and with chemical analysis line. This is done for clarity of the «substrate-coating» transition zone. In previous papers, reviewers have asked to do both methods, because for some it is clearer elemental mapping, for others ‒ the chemical analysis line. These are individual preferences and, more importantly, they do not impact the final result.

7 - row 207: (0, 1 and 3 wt.% B4C), if I understand correctly, it should be  (1  and 3 wt.% B4C),

Answer: Thanks for your observation. Yes, you are absolutely right it should be (1 and 3 wt.% B4C). We have corrected it in the manuscript.

8 - table 5: how authors calculated the reported parameters?

Answer: To calculate the period lattice (a) from a diffraction pattern, it is necessary to use the Wulf-Bragg condition, then the period can be calculated from the angular position of the line:

where H, K, L are the diffraction indices associated with crystallographic relationships: H = n·h, K = n·k, L = n·l, where n ‒ the reflection order.

Next, we find the volume (V) of the FCC lattice by raising the lattice period to the third power:

V = α3

9 - In my opinion ref 39 in row 292 is not pertinent

Answer: Thank you for your observation. We agree with this comment. This reference was used only for additional confirmation of the characteristic bands in the Raman spectra of boron carbides which were cited in the previous references [37-38]. The reference was removed from the manuscript.

10 - as regard figure 7: it is not clear in which way authors analysed the same spot in all the sample

Answer: Thank you for your observation. For Raman measurements we chose four different areas on the surface of each sample: spot 1 (grey areas), white areas on the coating (spot 2), dark-violet spots (3) and microdrops. These locations were found on the surface of all specimens under study (as can be seen here from the Figure 1). We have revised Fig. 7 and added optical images of the FeNiCr and FeNiCr+3wt.%B4C coatings to show different detection spots on the surface of each sample. We have measured about 5-7 Raman spectra at different points of each detection spot around the whole surface of each specimen. Also, we have revised the corresponding text in the manuscript.

11- in Raman analysis they found oxides: it is necessary to discuss this issue

Answer: Thank you for your observation. One of the features of Raman spectroscopy technique is that pure metals cannot show Raman bands as metals do not show the polarizability change during molecular vibration. Using Raman microscopy, we can only detect the metal oxides as a result of pure metal interaction of with oxygen. So, such 3d-metal as Ni, Fe and Cr used as the coating components are detected as NiO, Fe2O3, Cr2O3 etc. in the Raman spectra. We also have revised this in the text.

Comments on the Quality of English Language. Extensive editing of English language are require, some suggestions are reported in the revisions

Answer: We took into account and corrected all your comments, and also conducted extensive language editing of the manuscript with a native speaker.

Round 2

Author Response

  1. The title of paper is not correct. The term mechanostructural characterisation is not standard terminology.

Answer: Thank you for your comment. We have corrected mechanostructural in the title and text as mechano-structural (widely used term meaning mechanical and structural).

  1. Although this work was done by ex-situ methods. But authors have written in-situ method. It is not clear how authors are claiming for in-situ development of coating. Because B4Cwas directly added to the Fe Ni Cr medium entropy alloys.

Answer: Thanks for your observation. In-situ alloying here means that the B4C powder is not initially in the FeNiCr powder composition, but interacts with it in the process of laser exposure.

  1. The introduction section do not show any novelty of this research

Answer: Thanks for your observation. The novelty of the paper lies in the method of coating deposition, namely, pulsed laser cladding, as well as the study of in-situ alloying with boron carbide on mechanical characteristics of the composite FeNiCr-B4C coatings. We would like to note that the other two reviewers found the work interesting for readers and relevant.

  1. What is meaning of mechanostructural properties?

Answer: Thank you for your question. As mentioned above, mechano-structural is widely used term meaning mechanical and structural. That is, the mechanical and structural properties.

  1. The size of samples are 10 x 5 x5 It is written in line 91. Is it correct?

Answer: Yes, it is correct.

  1. In line 101, authors have written Fe Ni Cr-B4C samples. What is meaning whether coated or bulk?

Answer: Thanks for your question. Phrase «FeNiCr-B4C samples» mean coating + substrate.

  1. In Nano indentation testing different dwell times are taken such as 20, 10, and The time should be uniform for correct analysis of hardness and modulus of elasticity.

Answer: Thanks for your observation. This is a typo and the correct times are 20, 20 and 20 s.

  1. This paper does not show any significant relationship between microstructure and mechanical properties like hardness, modulus of elasticity and plasticity index.

Answer: The paper established the dependence of mechanical parameters such as hardness, modulus of elasticity and plasticity index on the concentration of boron carbide. In particular, the indentation microhardness of the FeNiCr coatings with 1 and 3 wt.% B4C rised by 16 and 38 %, and the contact elasticity modulus decreased by 4 and 10 %, respectively. In turn, this was one of the key goals of the study.

  1. Significant conclusion are not drawn from this research

Answer: All significant conclusions reflecting the goals of the study are given as follows:

  • The formed FeNiCr-B4C (0, 1 and 3 wt.% B4C) coatings are characterized by equiaxed coarse-grained structure. According to XRD analysis the coatings are homogeneous solid solutions and consist of a single FCC γ-phase, space group Fm-3m, independently of the B4C content. However, additional TEM analysis of the FeNiCr coating with 3 wt.% B4C showed that it is characterized by two-phase FCC structure consisting of the grains up to 1 µm in size and banded interlayers between the grains. The grains are clean; the density of dislocations inside the grains is low.
  • The cross-section of all obtained samples is characterized by average coating thickness of 400 ± 20 μm and sufficiently narrow, 100 ± 20 μm, “coating-substrate” transition zone. This indicates that the coating after 120 μm height consists only of laser-sintered equiatomic FeNiCr powder without mixing with the substrate material due to the double-pass laser cladding.
  • All obtained coatings have the same and insignificant number of defects: pores and cracks. This is probably due to the presence of 0.37 wt.% carbon and the coarse dispersion (50‒150 μm) of the initial FeNiCr powder. Thus, it can be concluded that there is no impact of in-situ alloying with B4C on the defect formation.
  • Raman spectroscopy analysis allowed to confirm the presence of B4C carbides in FeNiCr+1wt.%B4C and FeNiCr+3wt.%B4C coatings by detected peaks corresponding to amorphous carbon and peaks indicating the stretching of C-B-C chains in B4C carbides.
  • The mechanical characterization showed that in-situ alloying with 1 and 3 wt.% B4C is resulted in noticeable increase in microhardness, i.e. 16 and 38 %, with a slight decrease in ductility, i.e. 4 and 10 %, compared to the B4C-free FeNiCr coating. Thus, in-situ alloying with B4C can be considered as a promising method for hardening of laser deposited MEA FeNiCr coatings.

  1. What is the main difference between in-situ and ex situ process?

Answer: Thanks for your question. As mentioned above, in-situ alloying here means that the B4C powder is not initially in the FeNiCr powder composition, but interacts with it in the process of laser exposure.

  1. The main findings of this research work should be mentioned in conclusion

Answer: Thanks again for the same question. All the main findings of this research work are given in conclusion section as follows:

  • The formed FeNiCr-B4C (0, 1 and 3 wt.% B4C) coatings are characterized by equiaxed coarse-grained structure. According to XRD analysis the coatings are homogeneous solid solutions and consist of a single FCC γ-phase, space group Fm-3m, independently of the B4C content. However, additional TEM analysis of the FeNiCr coating with 3 wt.% B4C showed that it is characterized by two-phase FCC structure consisting of the grains up to 1 µm in size and banded interlayers between the grains. The grains are clean; the density of dislocations inside the grains is low.
  • The cross-section of all obtained samples is characterized by average coating thickness of 400 ± 20 μm and sufficiently narrow, 100 ± 20 μm, “coating-substrate” transition zone. This indicates that the coating after 120 μm height consists only of laser-sintered equiatomic FeNiCr powder without mixing with the substrate material due to the double-pass laser cladding.
  • All obtained coatings have the same and insignificant number of defects: pores and cracks. This is probably due to the presence of 0.37 wt.% carbon and the coarse dispersion (50‒150 μm) of the initial FeNiCr powder. Thus, it can be concluded that there is no impact of in-situ alloying with B4C on the defect formation.
  • Raman spectroscopy analysis allowed to confirm the presence of B4C carbides in FeNiCr+1wt.%B4C and FeNiCr+3wt.%B4C coatings by detected peaks corresponding to amorphous carbon and peaks indicating the stretching of C-B-C chains in B4C carbides.
  • The mechanical characterization showed that in-situ alloying with 1 and 3 wt.% B4C is resulted in noticeable increase in microhardness, i.e. 16 and 38 %, with a slight decrease in ductility, i.e. 4 and 10 %, compared to the B4C-free FeNiCr coating. Thus, in-situ alloying with B4C can be considered as a promising method for hardening of laser deposited MEA FeNiCr coatings.

  1. The line 312-314 is not indicating any meaningful

Answer: Thanks for your observation. If you mean «Table 5. Averaged peak positions of Raman bands detected at different surface spots on AISI 1040 steel substrate, B4C-free FeNiCr, FeNiCr+1wt.%B4C and FeNiCr+3wt.%B4C coatings then in the text you will find a detailed description of all the parameters indicated in this Table.

We are sincerely grateful for your comments. With you, our paper has become much better.

Reviewer 3 Report

Authors did not complete revisions; comments are in the attached file

Author Response

INTRODUCTION

1 - The introduction is not well written therefore it is hard to understand; for example:

row 40: ... to decrease in the plasticity of the latter...: what does it refer to?

Answer: Yes, you are completely right. We have removed the phrase «which in turn leads to decrease in the plasticity of the latter [7, since this phrase does not quite reflect the logic of the Introduction.

row 64:... melt  pool is formed from the substrate and deposited (powder or wire) materials: what does it means?

Answer: We agree with you, this text is really misleading. And since the above text, as in the first case, does not reflect the logic of the Introduction and, most importantly, it is not informative, we removed it. Thank you for your comment.

2 - real examples of application are not explicated

Answer: Thanks for your comment. We have added information about the possible use of our materials.

Answer 2: We apologize for this omission. Information added in row 53 as «According to [20], FeNiCr-based materials are commonly used as in-core components for nuclear light water reactors, as well as turbine disks and gas compressors.»

3 - before use acronym it is necessary to explicate the meaning (es: SLM row 62)

Answer: Yes, you are absolutely right. We took into account your comment and explicate this abbreviation in the text as «selective laser melting».

4 - the verb "to impact" should be used followed by "with"

Answer: Unfortunately, we did not find the verb "to impact" in the Introduction. If you meant this sentence «Thus, the research goal is to evaluate the impact of in-situ alloying with B4C on mechanostructural characteristics of laser deposited medium-entropy FeNiCr coatings.». The word «the impact» is used as a noun. But thanks a lot for your recommendation.

RESULTS

5 - TABLE 3: there is no carbon even if authors stated that an amount of carbon was presents in row 87

Answer: We agree, thanks for your observation. The main goal was to show the equiatomicity of the commercial powder. The EDS analysis also detected carbon in the amount declared by the manufacturer. We removed this table so as not to mislead the reader and left the information in the text.

6 - Figure 3. (a) Elemental mapping and (b) chemical analysis line of the cross-sectional “substrate-coating” zones of the B4C-free FeNiCr and FeNiCr+3wt.%B4C samples, respectively.  From this caption I understand: Elemental mapping is on B4C-free FeNiCr sample. If it is correct why a similar image of FeNiCr+3wt.%B4C is not reported?

Answer: Thank you for your comment. The main goal was to show the thickness of the transition (remelting) zone. This can be done both with elemental mapping and with chemical analysis line. This is done for clarity of the «substrate-coating» transition zone. In previous papers, reviewers have asked to do both methods, because for some it is clearer elemental mapping, for others ‒ the chemical analysis line. These are individual preferences and, more importantly, they do not impact the final result.

Answer 2: Thank you for your suggestion. We always treat the comments and suggestions of reviewers kindly and with trepidation. We would like to note once again that the goal of determining the elemental composition of the samples was achieved. In particular, the remelting and coating areas were determined. I think that you will agree that the absence of the figures you propose is not critical. In addition, there is currently no technical possibility to take the proposed SEM images.

7 - row 207: (0, 1 and 3 wt.% B4C), if I understand correctly, it should be  (1  and 3 wt.% B4C),

Answer: Thanks for your observation. Yes, you are absolutely right it should be (1 and 3 wt.% B4C). We have corrected it in the manuscript.

Answer 2: If we understood you correctly, then we have corrected it according to your comment as (1 and 3 wt.% B4C), please see the text below. Thus, we compare our FeNiCr coatings with 1 and 3 wt.% B4C with literature data.

8 - table 5: how authors calculated the reported parameters?

Answer: To calculate the period lattice (a) from a diffraction pattern, it is necessary to use the Wulf-Bragg condition, then the period can be calculated from the angular position of the line:

where H, K, L are the diffraction indices associated with crystallographic relationships: H = n·h, K = n·k, L = n·l, where n ‒ the reflection order.

Next, we find the volume (V) of the FCC lattice by raising the lattice period to the third power:

V = α3

Answer 2: Thanks again for your interest in lattice parameter calculations. Unfortunately, it is unethical to indicate such basic calculations as the period and volume of the lattice in the journals of the first quartile as Materials. Such information is public and has no scientific value. Please understand us correctly, we have already encountered this more than once when reviewing, and such calculations always cause a bad impression on readers.

9 - In my opinion ref 39 in row 292 is not pertinent

Answer: Thank you for your observation. We agree with this comment. This reference was used only for additional confirmation of the characteristic bands in the Raman spectra of boron carbides which were cited in the previous references [37-38]. The reference was removed from the manuscript.

10 - as regard figure 7: it is not clear in which way authors analysed the same spot in all the sample

Answer: Thank you for your observation. For Raman measurements we chose four different areas on the surface of each sample: spot 1 (grey areas), white areas on the coating (spot 2), dark-violet spots (3) and microdrops. These locations were found on the surface of all specimens under study (as can be seen here from the Figure 1). We have revised Fig. 7 and added optical images of the FeNiCr and FeNiCr+3wt.%B4C coatings to show different detection spots on the surface of each sample. We have measured about 5-7 Raman spectra at different points of each detection spot around the whole surface of each specimen. Also, we have revised the corresponding text in the manuscript.

Figure 1. Optical image of the a) FeNiCr, b) FeNiCr+1wt.%B4C and c) FeNiCr+3wt.%B4C coatings

11- in Raman analysis they found oxides: it is necessary to discuss this issue

Answer: Thank you for your observation. One of the features of Raman spectroscopy technique is that pure metals cannot show Raman bands as metals do not show the polarizability change during molecular vibration. Using Raman microscopy, we can only detect the metal oxides as a result of pure metal interaction of with oxygen. So, such 3d-metal as Ni, Fe and Cr used as the coating components are detected as NiO, Fe2O3, Cr2O3 etc. in the Raman spectra. We also have revised this in the text.

Answer 2: Thank you again for your interesting and rather curious question about oxides. However, your question was clear the first time and I will try to reformulate the answer of my colleagues. The specificity of Raman spectroscopy lies in the fact that it determines not pure elements, but their bonds (or chains), in particular, with oxygen, carbon, etc. For example, when determining the content of titanium or niobium in a coating, the Raman spectra show them in the form of TiO, NbO and etc. But we interpret them as pure elements. I kindly recommend to get acquainted with the basics of Raman spectroscopy. We are sincerely happy to share experiences with our colleagues and hope that our answer has become clearer.

Comments on the Quality of English Language. Extensive editing of English language are require, some suggestions are reported in the revisions
